# Expression, purification and initial characterization of human serum albumin domain I and its cysteine 34

**Martina Steglich**[1,2]**, Rodrigo Lombide**[1]**, Ignacio López**[3]**, Madelón Portela**[4,5]**, Martín Fló**[6,7]**, Mónica Marín**[3]**, Beatriz Alvarez**[1,2]**, Lucía Turell**[1,2]*

**1** Laboratorio de Enzimología, Facultad de Ciencias, Universidad de la República, Montevideo, Uruguay, **2** Center for Free Radical and Biomedical Research, Universidad de la República, Montevideo, Uruguay, **3** Sección Bioquímica, Facultad de Ciencias, Universidad de la República, Montevideo, Uruguay, **4** Unidad de Bioquímica y Proteómica Analíticas, Institut Pasteur de Montevideo and Instituto de Investigaciones Biológicas Clemente Estable, Montevideo, Uruguay, **5** Facultad de Ciencias, Universidad de la República, Montevideo, Uruguay, **6** Departamento de Inmunobiología, Facultad de Medicina, Universidad de la República, Montevideo, Uruguay, **7** Laboratorio de Inmunobiología, Institut Pasteur de Montevideo, Montevideo, Uruguay

* lturell@fcien.edu.uy

**Data Availability Statement:** All relevant data are within the manuscript and its Supporting Information files.

## Abstract

Human serum albumin presents in its primary structure only one free cysteine (Cys34) which constitutes the most abundant thiol of plasma. An antioxidant role can be attributed to this thiol, which is located in domain I of the protein. Herein we expressed domain I as a secretion protein using the yeast *Pichia pastoris*. In the initial step of ammonium sulfate precipitation, a brown pigment co-precipitated with domain I. Three chromatographic methods were evaluated, aiming to purify domain I from the pigment and other contaminants. Purification was achieved by cation exchange chromatography. The protein behaved as a non-covalent dimer. The primary sequence of domain I and the possibility of reducing Cys34 to the thiol state while avoiding the reduction of internal disulfides were confirmed by mass spectrometry. The reactivity of the thiol towards the disulfide 5,5´-dithiobis(2-nitrobenzoate) was studied and compared to that of full-length albumin. A ~24-fold increase in the rate constant was observed for domain I with respect to the entire protein. These results open the door to further characterization of the Cys34 thiol and its oxidized derivatives.

## Introduction

Human serum albumin (HSA) is the most abundant protein in plasma (0.6 mM), comprising about 60% of total plasma proteins. It is synthesized in the liver as a pre-pro-protein with two additional segments, the prepeptide and the propeptide. The prepeptide (18 aminoacids) constitutes the secretion signal and leads HSA out of the hepatocyte, while the propeptide (6 aminoacids) is of unknown function. Both segments are located in the N-terminal portion of the protein and are cleaved before protein secretion, resulting in mature HSA. This non-glycosylated protein is the principal macromolecular anion in plasma, with 19 negative charges at pH

**Funding:** This work was supported by grants from Agencia Nacional de Investigación (ANII, Uruguay, PR_FCE_2009_1988) to L.T. and Comisión Sectorial de Investigación Científica (CSIC, Uruguay) to B.A. and L.T., and fellowships from Agencia Nacional de Investigación e innovación (ANII, Uruguay) to R.L. and Comisión Académica de Posgrado, Universidad de la República (CAP, UDELAR, Uruguay) to M.S.

**Competing interests:** The authors have declared that no competing interests exist.

7.4. Several functions are attributed to HSA. Some of them include the maintenance of colloid osmotic pressure and the binding and transport of a wide variety of ligands such as fatty acids, bilirubin, hormones, metal ions and drugs [1].

HSA is a monomeric protein consisting of three homologous domains (domains I, II and III). Among its 585 aminoacids it possesses 35 cysteines that form 17 disulfide bonds. Domain I (Dom I) comprises residues 1 to 195 and stands out for having the only free cysteine of the protein, Cys34 [1, 2]. In plasma, the thiol group of Cys34 is the main responsible for the anti-oxidant properties of the protein, as it is the most abundant reduced thiol and an important target for oxidizing reactive species [3]. Reduced Cys34 can react with disulfides in thiol exchange reactions [4, 5]. It can also react with two-electron oxidants, such as hydrogen peroxide, hypochlorous acid and peroxynitrite, to form a sulfenic acid [6–11]. This intermediate can be further oxidized to irreversible forms such as sulfinic and sulfonic acids, which are not reducible by common reducing agents such as dithiothreitol (DTT) or 2-mercaptoethanol. Sulfenic acid can also react with low molecular weight thiols to form mixed disulfides or decay spontaneously to an uncharacterized species [9].

The use of *Pichia pastoris* as a system for the expression of heterologous proteins has increased over the years, as it presents several advantages compared to other systems. As an eukaryotic organism, *P. pastoris* is able to make posttranslational modifications like disulfide bond formation, glycosylation and proteolytic cleavage. Moreover, it is possible to secrete recombinant proteins to the culture medium, which constitutes a purification advantage as the levels of endogenous proteins secreted by *P. pastoris* are low. The ability to integrate the expression vector to the yeast genome in one or multiple copies is also an important feature of this system, since the number of copies is related with the expression level of the recombinant protein [12, 13]. There are several reports of the expression of HSA in *P. pastoris* [14–20] some of which use its native signal, the prepeptide, to express it as a secretion protein [17, 18].

To further investigate the properties of Cys34 and its oxidation derivatives, we expressed DomI using *P. pastoris* and purified it from the culture medium. The oligomeric state of the protein, the reducibility and the reactivity of Cys34 were assessed.

## Materials and methods

### Materials

All reagents were from Sigma or PanReac-Applichem unless otherwise specified. *P. pastoris* electroporation was performed on an electroporator Gene Pulser XCell, Bio-Rad. Absorbance determinations were made on a Varian Cary 50 spectrophotometer and chromatography experiments on a FPLC AKTA-Prime Plus. HiTrap Blue HP, HiPrep CM FF 16/60 and Superdex 75 10/300 columns were from GE Healthcare. Matrix assisted laser desorption ionization time of flight (MALDI-TOF) mass spectrometry (MS) experiments were performed on a MALDI-TOF/TOF 4800 AB-Sciex instrument. Proteomics grade trypsin from Sigma or Sequencing grade modified trypsin from Promega were used in MS experiments. A stock solution of 5,5′-dithiobis(2-nitrobenzoate) (DTNB, 20 mM) was prepared in ethanol and working solutions were prepared in phosphate (20 mM, pH 7.4) buffer. HSA (Sigma) was delipidated by activated charcoal treatment [21], and protein concentration was determined from the absorbance at 279 nm ($\varepsilon$ = 0.531 (g/L)$^{-1}$ cm$^{-1}$, MW = 66438 Da) [1].

### Cloning of HSAwt and DomI

Total RNA from HepG2 cells was isolated using the RNeasy mini kit (QIAGEN) following manufacturer's instructions. Retrotranscription was performed using 2.5 µg of RNA, a specific HSA oligonucleotide (ATAAGCCTAAGGCAGCTTGACTGG) and Superscript II reverse

transcriptase (Invitrogen). The full-coding sequence of HSA was PCR- amplified with U-Taq (SBS GeneTech) and sense (CGCGAATTCATGAAGTGGGTAACC) and antisense (CGCCTCGAGTTATAAGCCTAAGGCAGC) primers containing EcoRI and XhoI restriction sites (underlined), respectively. The amplified product was cloned into a pGEM vector (Promega), sequenced and subcloned into pPICZA vector (Invitrogen). The propeptide sequence (AGGGGTGTGTTTCGTCGA) was deleted by PCR using primers flanking the target sequence (reverse GGAATAAGCCGAGCTAAAGAGAAAAAGAAGGG and forward GATGCACACAAGAGTGAGGTTGCTCATCGG) previously phosphorylated with T4 polynucleotide kinase T4-PNK (Thermo Scientific). T4 ligase (Thermo Scientific) was used to blunt end ligate the PCR product. Domain I (DomI) coding region was obtained from the last by PCR amplification using KAPA HiFi polymerase (Kapa Biosystems) and specific sense (CGCGAATTCATGAAGTGGGTAACC) and antisense (CGCCTCGAGTTATTTGGCAGACGAAGCCTT) primers containing EcoRI and XhoI restriction sites (underlined), respectively. Then it was subcloned back into pPICZA (pPICZA-DomI).

## *Pichia pastoris* transformation

The vector pPICZA-DomI was linearized with the restriction enzyme PmeI and used to transform the *P. pastoris* strain GS115, His⁻, Mut⁺ (Invitrogen) by electroporation following the EasySelect *Pichia* Expression Kit recommendations (Invitrogen). Transformants were selected in YPDS agar (1% yeast extract, 2% peptone, 2% dextrose, 1 M sorbitol, 2% agar) with Zeocin (100 μg/mL) after incubation for 6 days at 30˚C. To isolate multicopy recombinants, all transformants were plated in YPDS agar with a higher Zeocin concentration (700 μg/mL) and incubated for 48 h at 30˚C.

## DomI expression

Transformed *P. pastoris* cells were grown for 2–3 days in YPD agar (1% yeast extract, 2% peptone, 2% dextrose, 2% agar) containing Zeocin (100 μg/mL). Liquid YPD cultures (5 mL) were started from single colonies, grown over 7–8 h and then used to inoculate 25 mL BMGY (1% yeast extract, 2% peptone, 100 mM potassium phosphate pH 6.0, 1.34% yeast nitrogen base, 4.5 x $10^{-5}$% biotin, 1% glycerol) to an initial $A_{600}$ of 0.02. After reaching an $A_{600}$ of 2–6, cultures were centrifuged and cells were resuspended in 100 mL BMMY (1% yeast extract, 2% peptone, 100 mM potassium phosphate pH 6.0, 1.34% yeast nitrogen base, 4.5 x $10^{-5}$% biotin, 0.5% methanol) for methanol induction. Methanol (0.5%) was added every 24 h and aliquots were taken over time to evaluate protein expression by SDS-PAGE. All incubations were performed at 30˚C under agitation.

## Protein precipitation with ammonium sulfate

After 96 h of methanol induction, the culture was centrifuged (4000 g, 10 min, room temperature), and proteins in the supernatant were precipitated overnight by the addition of ammonium sulfate (70% saturation), at pH 4 and 4˚C. After centrifugation (12000 g, 30 min, 4˚C) the pellet was resuspended in water and dialyzed (cut-off 14 kDa) for 24 h against water and then for 24 h against phosphate buffer (20 mM, pH 7.4). The resulting sample is called "dialyzed sample".

## Affinity chromatography

The dialyzed sample was applied to a Blue Sepharose 6 Fast Flow column previously equilibrated with phosphate buffer (20 mM, pH 7) and washed with the same buffer at a flow rate of

3 mL/min. Elution was performed with NaCl-containing phosphate buffer (20 mM, pH 7, 2 M NaCl).

## Batch anion exchange chromatography

DEAE Sepharose resin was used. Several equilibration buffers were evaluated: ammonium acetate (20 mM, pH 5.2), urea-containing ammonium acetate (20 mM, pH 4.0, 6 M urea), Tris (50 mM, pH 7.4), and urea-containing formic acid (100 mM, pH 3.0, 6 M urea). In all cases, the dialyzed sample buffer was changed to the corresponding equilibration buffer using ultrafiltration devices. After loading the sample, three washes with equilibration buffer were performed. Elution was performed manually with a gradient of NaCl (0.05–1 M).

## Column cation exchange chromatography

The buffer of the dialyzed sample was changed to the equilibration buffer (20 mM sodium acetate, pH 5.0) by ultrafiltration. Then, the sample was applied to a HiPrep CM Fast Flow column previously equilibrated with the corresponding buffer. The column was washed and the elution was performed with the same buffer containing 1 M NaCl at a flow rate of 3 mL/min.

## Protein quantification

Three methods were evaluated to determine DomI concentration. Bradford, bicinchoninic acid assay (BCA) and 280 nm absorbance determinations were performed [22, 23]. For Bradford and BCA measurements, calibration curves were done with delipidated HSA. For the BCA assay, samples were heated at 60˚C during 15 min for color development. Controls with the sample in the absence of BCA or Bradford reagents were included. For protein quantification by 280 nm absorbance, an extinction coefficient of $\varepsilon = 11055$ M$^{-1}$ cm$^{-1}$ was determined from the primary sequence of DomI using the ProtParam tool (ExPASy, Bioinformatics Resource Portal).

## Analytical size exclusion chromatography

Purified DomI was diluted to a concentration of 90 μg/mL in NaCl-containing Tris buffer (50 mM, pH 8, 100 mM NaCl) and 100 μL were applied to a Superdex 75 10/300 column, previously equilibrated with the same buffer at a flow rate of 0.8 mL/min and calibrated using known proteins. Absorbance was registered at 280 and 215 nm.

## SDS-PAGE

Electrophoresis was performed in 15% polyacrylamide gels containing sodium dodecyl sulfate (SDS). Samples were prepared under reducing and non-reducing conditions depending on the presence or absence of 2-mercaptoethanol. Proteins were visualized with colloidal Coomassie blue staining.

## Reductive alkylation of DomI and mass spectrometry analysis

Purified DomI (30 μM) was exposed to guanidinium chloride (6 M) and DTT (2 mM). The sample was purged with argon and incubated for 2 h at 37˚C. Thiols were carboxymethylated with iodoacetic acid (10 mM) for 1 h in the dark, followed by the addition of 2-mercaptoethanol (100 mM). The sample was washed and concentrated by ultrafiltration using ammonium bicarbonate (0.1 M), and incubated overnight at 37˚C with trypsin (DomI to trypsin ratio 50:1). MALDI-TOF mass spectra of the digestion mixture were acquired using a matrix solution of α-cyano-4-hydroxycinnamic acid in 60% acetonitrile and 0.1% trifluoroacetic acid. The

calibration was performed with a mixture of standard peptides (Applied Biosystems). MS and MS/MS data were acquired in positive reflector mode.

## Differential thiol labeling and mass spectrometry analysis

Purified DomI (25 μM) was reduced by incubation with 2-mercaptoethanol (10 mM), overnight at 4˚C [24]. Iodoacetamide (50 mM) was added and the mixture was incubated for 1 h in the dark, at room temperature. The sample was washed by ultrafiltration with ammonium bicarbonate (0.1 M) and was then treated with DTT (2 mM) for 2 h at 37˚C. Iodoacetic acid (10 mM) was added and the mixture was incubated for 1 h in the dark at room temperature. Finally, the sample was washed with ammonium bicarbonate (0.1 M) and incubated overnight at 37˚C with trypsin (DomI to trypsin ratio 50:1). MS and MS/MS data were obtained as described above. An internal calibration with DomI peptides was performed, with an error less than 0.08 Da.

## Thiol reduction, quantification and reactivity with DTNB

Both HSA (1.5 mM) and purified DomI (44 μM) were reduced by incubation with 2-mercaptoethanol (10 mM), 30 min at room temperature or overnight at 4˚C [24, 25]. Excess reductant was eliminated by gel filtration against phosphate buffer (20 mM, pH 7.4). Reduced DomI (11.5 μM) was mixed with 50–200 μM DTNB, while reduced HSA (20 μM) was mixed with 0.5–2.5 mM DTNB. In both cases the formation of thionitrobenzoate (TNB) was followed by the increase in absorbance at 412 nm ($\varepsilon = 14150$ $M^{-1}$ $cm^{-1}$) [26]. Exponential plus straight line equations were fitted to the time courses and the pseudo-first order rate constants ($k_{obs}$) were determined. Second order rate constants were obtained from the $k_{obs}$ versus DTNB concentration secondary plots. Controls were performed with only reduced DomI (11.5 μM), only DTNB (200 μM) or with the mixture of non-reduced DomI (11.5 μM) and DTNB (200 μM).

# Results

## DomI expression and purification

The construct for DomI (pPICZA-DomI) expression was obtained. The sequence of DomI construct codes for aminoacids 1 to 195 of HSA, plus the 18 prepeptide aminoacids, and contained two synonymous mutations with respect to the reference sequence (NM_000477.6). The pPICZA-DomI vector was used to transform *Pichia pastoris* GS115 cells and the transformants were selected with Zeocin (100 μg/mL). Five of the fastest growing clones on a medium containing higher Zeocin concentration (700 μg/mL) were evaluated for DomI expression. The clone with the highest expression level of DomI was selected.

During the 96 h induction with methanol, aliquots were taken every 24 h to evaluate protein expression by SDS-PAGE (Fig 1A). After 24 h of induction, a band with an apparent molecular weight consistent with Dom I (22371 Da, ProtParam, Expasy) was observed. The intensity of this band increased with the induction time. Noteworthy, DomI was the major protein secreted by the yeast to the culture medium.

Proteins were purified from the culture medium by ammonium sulfate precipitation and the pellet was resuspended in distilled water. Unexpectedly, the resuspended pellet had a brown color. The pigment did not bind to Coomassie blue when SDS-PAGE was assessed, suggesting it did not consist of protein. This was confirmed by MALDI-TOF MS/MS. After 24 h of dialysis against water and 24 h against buffer, part of the pigment was observed in the dialysate but the protein mixture was still brown (Fig 1B).

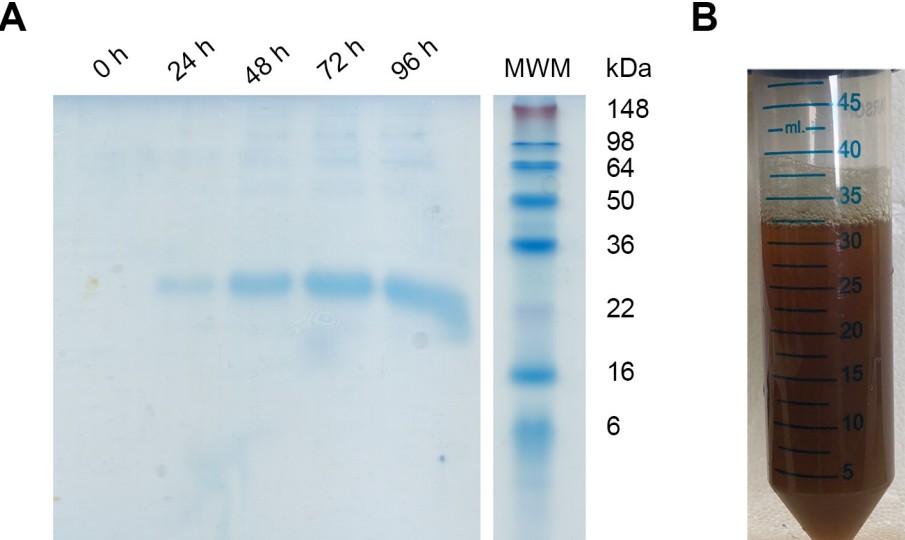

**Fig 1. DomI expression and purification from the culture medium.** (A) Reducing SDS-PAGE of the culture supernatant. Samples were taken every 24 h during the 96 h of methanol induction. After centrifugation to remove cells, supernatants were diluted in 4 x sample buffer and 15 μL were loaded. Empty lanes were cropped. (B) Picture of the sample obtained after culture centrifugation, protein precipitation with ammonium sulfate, pellet resuspension and dialysis.

In order to purify DomI from the pigment and from the contaminant proteins, different approaches were attempted. First, Blue Sepharose affinity chromatography [27, 28] was assayed. This method is widely used to purify HSA [18–20, 29], as this protein interacts with the Cibacron F3G-A groups of the resin. In the case of DomI, only a minimal fraction was adsorbed to the resin and most of the protein eluted in the flowthrough along with the pigment and contaminant proteins (not shown). Second, ion exchange chromatography was tried. The isoelectric point of DomI was calculated from its primary sequence using the ProtParam tool. A value of 5.3 was obtained, which is in agreement with the previously reported value of 5.2 [14]. Anion exchange chromatography (DEAE Sepharose) in batch was performed. Different buffers were evaluated (pH 3.0–7.4) and the elution was always performed by an increase in ionic strength, adding NaCl. Despite the different conditions that were evaluated, it was not possible to separate the protein from the pigment (not shown). Furthermore, part of the pigment stayed adsorbed to the resin even after addition of 1 M NaCl hindering the regeneration of the resin. This was indicative that the pigment was negatively charged.

Third, cation exchange chromatography was performed using a CM Sepharose column. The starting buffer was sodium acetate (20 mM, pH 5.0) and the elution buffer contained also 1 M NaCl. Two well defined peaks were obtained in the chromatogram; the first one corresponded to the flowthrough, while the second one, to the bound fraction that was eluted by a jump in ionic strength (Fig 2A). As revealed by SDS-PAGE, the flowthrough contained mostly contaminant proteins and DomI, while the bound fraction contained mostly DomI (Fig 2B). The UV-Vis spectra of the three samples showed that the bound fraction eluted from the column had a maximum absorbance at 276 nm and a decrease in absorbance at 300–450 nm, compared to the flowthrough (Fig 2C). Also, the sample looked colorless instead of brown. These observations suggest that the bound fraction presented a lower content of pigment than the dialyzed sample and the flowthrough. As most of the pigment and contaminant proteins were separated from DomI, this method was used to purify DomI for the experiments that follow.

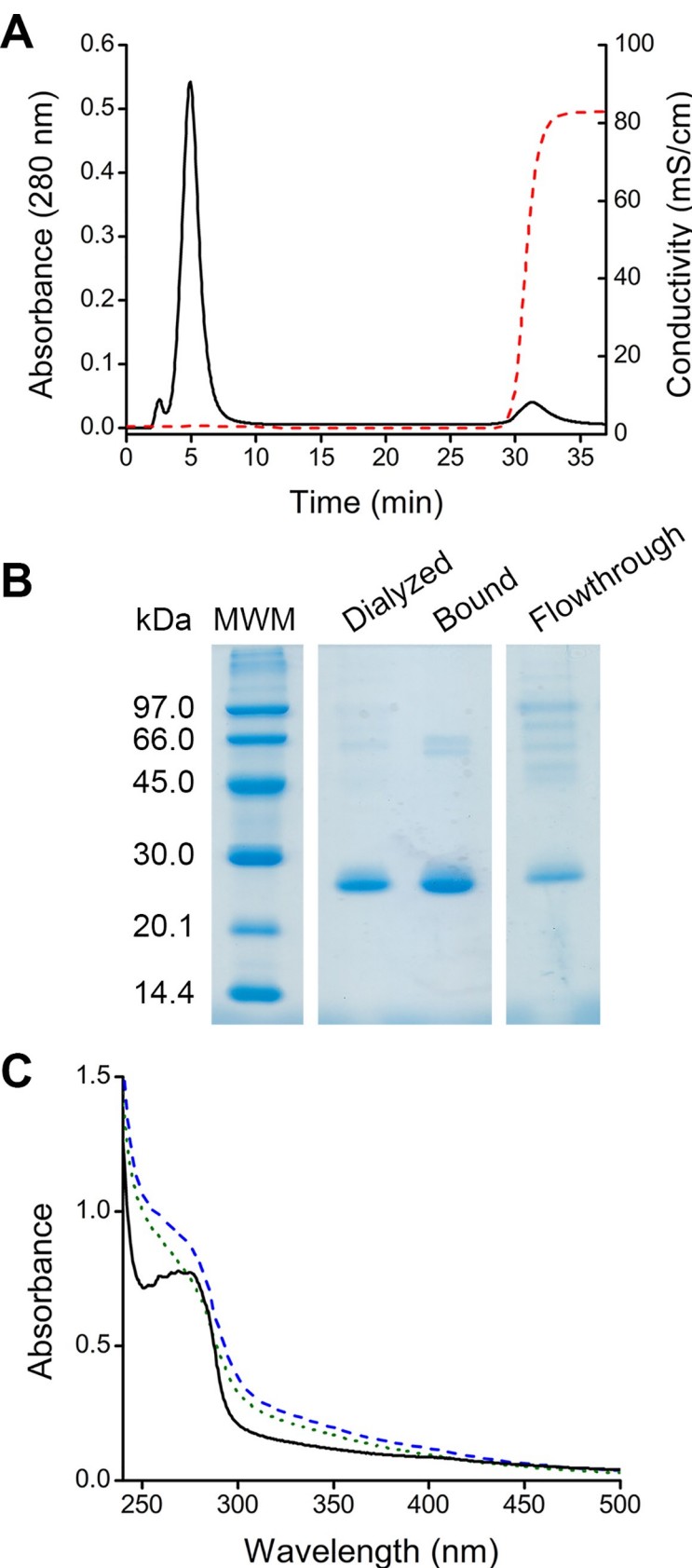

**Fig 2. DomI purification by cation exchange chromatography.** (A) The dialyzed sample was loaded into a HiPrep CM FF column, previously equilibrated with sodium acetate buffer (20 mM, pH 5.0), at 3 mL/min flow rate. Elution was performed by a jump in ionic strength (acetate buffer 20 mM, pH 5.0, NaCl 1M). Absorbance at 280 nm (black solid trace) and conductivity (red dashed trace) were measured. (B) SDS-PAGE analysis. The flowthrough and the bound fraction eluted from the column were concentrated using ultrafiltration devices before loading. Empty lanes were cropped. (C) UV-Vis spectra of the different fractions, dialyzed sample (blue dashed trace), flowthrough (green dotted trace) and bound fraction eluted from the column (black solid trace). Dilutions with similar 280 nm absorbance were prepared.

Regarding protein quantification, the presence of the pigment, mainly in the dialyzed sample, did not allow the direct measurement of the absorbance at 280 nm. Also, interferences due to the pigment were observed with the BCA method (data not shown). Thus, protein concentration was determined using the Bradford assay both in the dialyzed and purified DomI samples. An estimated yield of 0.1 mg of DomI per mL of culture was obtained.

## DomI oligomeric state

To characterize the oligomeric state of DomI, analytical size exclusion chromatography was performed. Purified DomI was loaded on a Superdex 75 column previously equilibrated with NaCl-containing Tris buffer (50 mM, pH 8, 100 mM NaCl). Considering the elution volume, a molecular weight of 45 kDa was calculated for DomI indicative that, in these experimental conditions, the protein was a dimer (Fig 3). With the concentration of purified DomI loaded (90 µg/mL) no other peaks were observed.

The nature of the dimer was evaluated by SDS-PAGE. Electrophoretic migration of DomI was compared under reducing and non-reducing conditions (not shown). Both samples migrated similarly, meaning that the dimer was not covalent and that no disulfide bonds were formed between monomers.

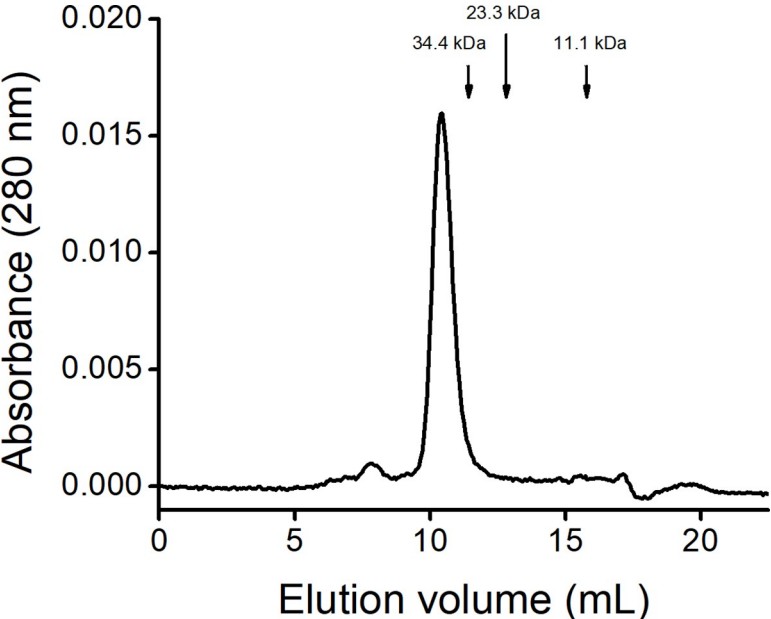

**Fig 3. DomI oligomeric state.** Purified DomI (100 µL, 90 µg/mL) was loaded on a Superdex 75 10/300 column previously equilibrated with Tris (50 mM, pH 8.0, 100 mM NaCl) at 0.8 mL/min and absorbance at 280 nm was registered. Arrows indicate the elution volume of the proteins used in the calibration.

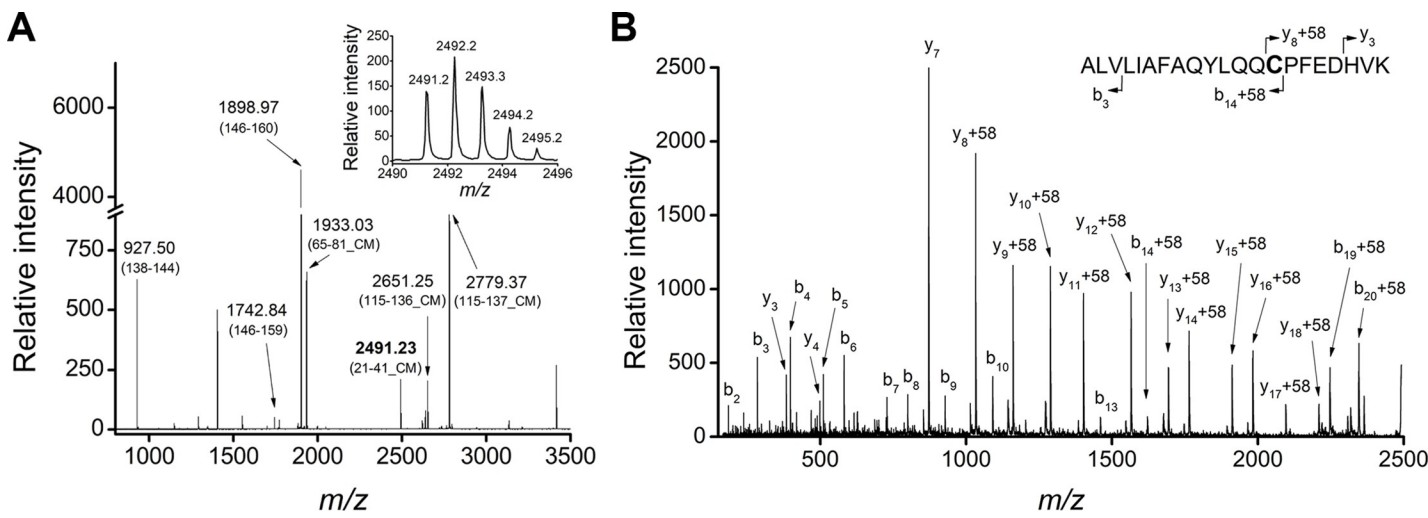

**Fig 4. MALDI-TOF MS analysis of DomI after reductive alkylation and trypsinization.** Purified DomI was reduced under denaturing conditions, alkylated with iodoacetic acid and trypsinized. (A) MALDI-TOF MS of DomI tryptic fragments. The peak with *m/z* 2491.23 (in bold) corresponds to the Cys34-containing peptide, carboxymethylated (CM, + 58). Inset: *m/z* 2491.23 peptide isotopic distribution. (B) MALDI-TOF MS/MS of the parent ion at *m/z* 2491.23 showing the identified *b* and *y* fragments.

## Analysis of the primary sequence of Dom I and oxidation state of Cys34

Purified DomI was reduced with DTT under denaturing conditions and was then treated with iodoacetic acid. The protein was trypsinized and analyzed by MALDI-TOF MS (Fig 4). Several peaks were identified as DomI fragments, based on the monoisotopic *m/z* value. These fragments were assigned to the DomI sequence, obtaining a 71% of coverage. No signals corresponding to the prepeptide were found on the MS, even considering missed cleavages in the primary sequence. This suggested that the proteolytic processing of this signal peptide was successful.

Because DomI was expressed during four days with agitation at 30˚C, the possibility that the Cys34 thiol became irreversibly oxidized (*e.g.* to sulfinic or sulfonic acids) was a major concern. Nevertheless, a signal consistent with the expected mass of the Cys34-containing peptide (2433.26 Da), carboxymethylated (+ 58.03 Da), was observed at *m/z* 2491.23 (Fig 4A). The MS/MS of the *m/z* 2491.23 signal was obtained, in order to confirm the identity of the peptide and the position of the modification. Several *y* and *b* ions were identified as belonging to the Cys34 peptide, confirming that the carboxymethylated residue was Cys34 (Fig 4B). These results confirmed that the thiol of Cys34 is reducible under reductive alkylation conditions and thus, it does not become irreversibly oxidized during protein expression and purification.

## MALDI-TOF analysis of selectively reduced Cys34

Before characterizing the reactivity of Cys34, it was necessary to reduce it selectively avoiding the reduction of the five intraprotein disulfides. For that, reduction conditions reported for HSA were used [24]. Purified DomI was first reduced with 2-mercaptoethanol (10 mM, 4˚C, overnight) and then treated with iodoacetamide. After that, DTT was added to the mixture, to reduce all the disulfides in the protein. The resulting thiols were blocked with iodoacetic acid, the sample was trypsinized and analyzed by MALDI-TOF MS (Fig 5). The modification of a thiol by iodoacetamide (carbamidomethylation) or iodoacetic acid (carboxymethylation) renders a mass increase of +57.05 or +58.03 Da, respectively.

Several peaks were identified as DomI fragments, reaching a 90% coverage. A signal of *m/z* 2490.33 was observed, which corresponds to the Cys34-containing peptide (2433.26 Da),

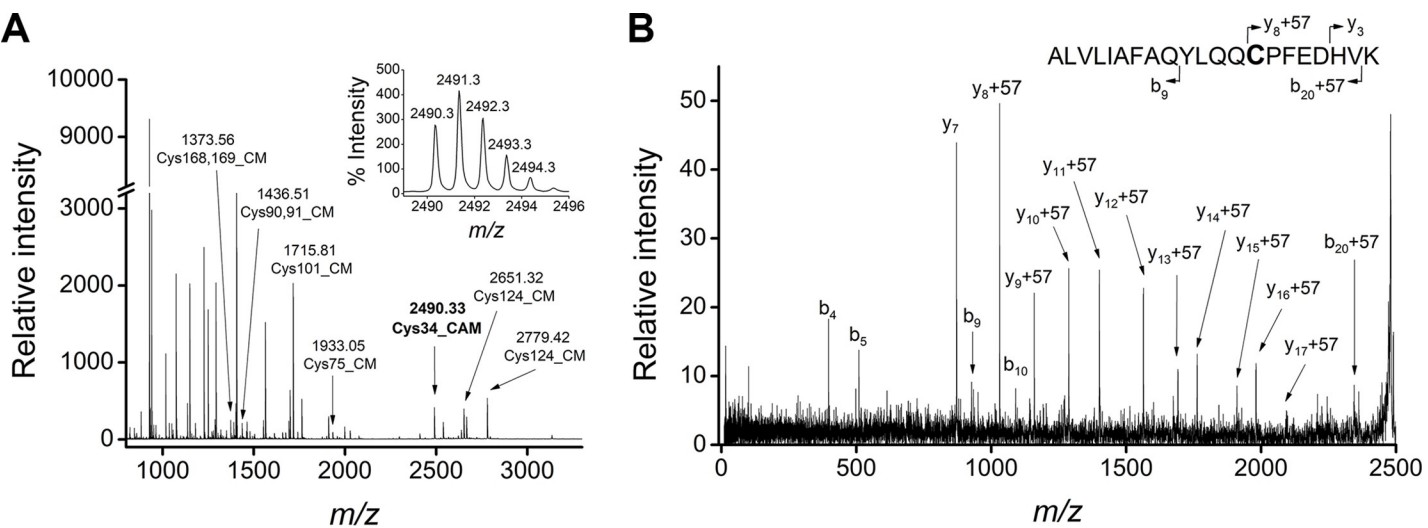

**Fig 5. Selective reduction of Cys34 and MALDI-TOF MS analysis.** Purified DomI was incubated with 2-mercaptoethanol and treated with iodoacetamide. Then, DTT was added to the mixture followed by iodoacetic acid treatment and trypsinization. (A) MALDI-TOF MS of DomI tryptic fragments. Carboxymethylated (CM) and carbamidomethylated (CAM) Cys-containing peptides are shown. Inset: $m/z$ 2490.33 peptide isotopic distribution. (B) MALDI-TOF MS/MS of the parent ion at $m/z$ 2490.33 showing the $b$ and $y$ fragments identified.

carbamidomethylated (Fig 5A). The isotopic distribution pattern of this signal reflects homogeneous labeling (Fig 5A, inset), meaning that there was no labeling by iodoacetic acid. Moreover, the MS/MS of the peptide was consistent with Cys34 being the carbamidomethylated residue (Fig 5B).

Regarding the modification of other cysteine residues, several of them were found carboxymethylated (+58.03 Da), meaning that they were reduced by DTT (Fig 5A). None of these peptides were found carbamidomethylated, discarding the possibility that cysteine residues different than Cys34 were reduced by 2-mercaptoethanol.

## DomI reduction and reactivity with DTNB

In order to evaluate the reactivity of DomI Cys34 and compare it to that of HSA, the reaction with DTNB was studied. This reporter disulfide has been previously used with HSA [9, 24, 25, 30] and allows to monitor the reaction through the absorbance of the product, TNB (Fig 6A). Purified DomI was incubated with 2-mercaptoethanol (10 mM, 30 min, room temperature), excess reductant was eliminated by gel filtration and the resulting reduced DomI (11.5 μM) was mixed with increasing concentrations of DTNB in pseudo-first order excess (50–200 μM). For comparison, the experiment was also performed with reduced HSA (20 μM). The time courses showed that DTNB reacted faster with DomI than with HSA (Fig 6B). Exponential plus straight line equations were fitted to the data. The first phase (exponential fit) corresponds to the reaction between DTNB and Cys34, and the second phase (straight line fit) can be explained in terms of secondary reactions such as DTNB dismutation [24]. The amplitude and pseudo-first order rate constant ($k_{obs}$) were obtained. Then, $k_{obs}$ were plotted against DTNB concentration (Fig 6C). From this secondary plot, second order rate constants of 110.3 $M^{-1}$ $s^{-1}$ and 4.6 $M^{-1}$ $s^{-1}$ were determined for DomI and HSA, respectively (20 mM phosphate, pH 7.4, 25°C). Thus, it can be concluded that DomI Cys34 reacted with DTNB ~24 times faster than HSA Cys34. A control of non-reduced DomI (11.5 μM) and DTNB (200 μM) was also performed and no reaction was observed (not shown).

The concentration of reduced thiols in DomI was determined from the amplitude of the time courses of the reaction with DTNB. To account for the fact that the first ~10 seconds

were missed, the amplitude obtained from the fit was corrected considering as the initial absorbance, the sum of the absorbance of the controls (only DTNB and only DomI) (Fig 6B). Taking into account the total protein concentration (DomI) used in the experiment, a thiol/DomI ratio of 0.20 was obtained. This suggests that 20% of DomI Cys34 was reduced after 2-mercaptoethanol incubation, 30 min at room temperature. Similar results were obtained with overnight reduction at 4˚C (not shown).

Given the low thiol/DomI ratio obtained by reduction with 2-mercaptoethanol, we searched for the Cys34-containing peptide (2433.26 Da) in other oxidation states. None of the following modifications were found in the MS analysis: sulfenic (+ 16 Da, $m/z$ 2449.26), sulfinic (+ 32 Da, $m/z$ 2465.26) and sulfonic (+ 48 Da, $m/z$ 2481.26) acids, sulfenamide (- 2.01 Da, $m/z$ 2431.25), sulfinamide (+ 13.98 Da, $m/z$ 2447.24), sulfonamide (+ 29.98 Da, $m/z$ 2463.24), dehydroalanine (- 34.08 Da, $m/z$ 2399.18) and the mixed disulfide with 2-mercaptoethanol (+ 76.12 Da, $m/z$ 2509.38). This suggests that Cys34 may be covalently modified by compounds of unknown identity, in a way that it is not reducible by 2-mercaptoethanol or DTT.

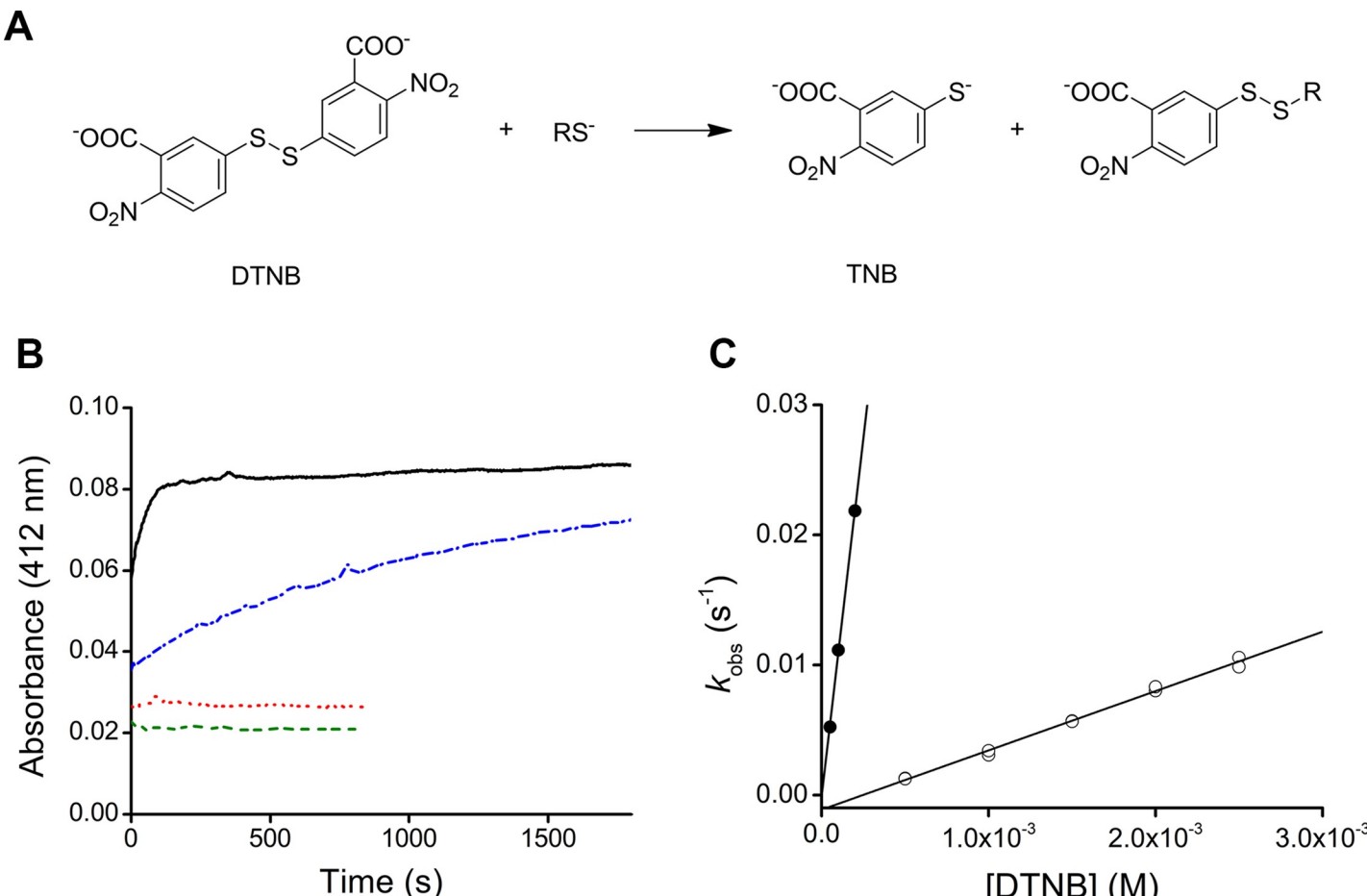

**Fig 6. Reactivity between Cys34 and DTNB.** (A) Scheme of the reaction between DTNB and a thiol. (B) DTNB (0.2 mM) was mixed with reduced DomI (11.5 μM, black solid trace) or reduced HSA (10 μM, blue dash-dotted trace) in phosphate buffer (20 mM, pH 7.4), and absorbance at 412 nm was recorded at 25˚C. Exponential plus straight line equations (Abs = amplitude (1 −exp (-$k_{obs}t$) + slope $t$ + offset)) were fitted to the data to determine the amplitude and $k_{obs}$. Controls with DTNB alone (red dotted trace) or DomI alone (green dashed trace) were also performed. (C) Secondary plot for the reaction of DTNB and DomI (11.5 μM, black circles) or HSA (20 μM, open circles). For both proteins, the second order rate constants of the reaction with DTNB were obtained from the slope of the linear fit.

## Discussion

DomI from HSA was successfully expressed and secreted to the culture medium by the yeast *P. pastoris*. Surprisingly a brown non-proteic pigment was co-expressed with our protein. Even though *P. pastoris* is a widely used system for the expression of recombinant proteins, there are only very few reports about pigments like the one observed here [31–34]. According to the scarce literature, these pigments have been observed when working with GS115 and X-33 strains of *P. pastoris*, while using different culture media for the expression [31, 33, 34]. It has been suggested that these pigments could be synthesized by the yeast during growth, as a consequence of the overexpression of the heterologous protein or the methanol induction conditions, which are both common factors in all the reported cases [31, 33, 34]. It has been shown that these pigments interact non-covalently with the recombinant protein and that they vary in composition, size and charge, making it difficult to remove them completely. In all the previous reports, more than one technique was necessary to achieve purification, while in general a small amount of pigment remained [31–34].

In this work, three chromatographic methods were evaluated to purify DomI from the pigment and contaminant proteins. Regarding affinity chromatography, DomI only marginally interacted with the resin making this method unsuitable for its purification. Dockal *et al* [14], previously showed that DomI interacts with the Cibacron groups, although to a lesser extent than domains II and III. This is in agreement with other reports that suggest that the interaction with Cibacron groups occurs mainly with domains II and III [28, 35]. Anion exchange chromatography was neither useful for DomI purification. Furthermore, the pigment stayed adsorbed to the resin, suggesting that it was negatively charged. Indeed, taking advantage of this, we finally managed to remove the pigment by cation exchange chromatography. The resulting DomI appeared colorless and presented a maximum absorbance at 276 nm. It is worth noting that DomI contains no tryptophan residues but seven tyrosine residues and thirteen phenylalanine residues. According to the analytical size exclusion chromatography, DomI behaves as a non-covalent dimer in the conditions tested. This is different from the monomer reported by Dockal *et al* [14], which can be explained by the ionic strength of the buffers used in each chromatography (0.37 M in Dockal's report and 0.13 M in our case), as the concentration of loaded sample was very similar (90 and 100 μg/mL).

High sequence coverage was obtained through MS experiments, confirming the primary sequence of the protein. The absence of the prepeptide in the final product (DomI) was evidenced, indicative of a correct proteolytic cleavage by the yeast, which is in agreement with previous reports [17, 18].

DomI Cys34 was reducible under conditions like those used to reduce HSA Cys34 [9, 24, 25]. Even after 96 h incubation under agitation, a thiol/DomI ratio of 0.2 was obtained. A question that remains to be answered is how is the rest of Cys34 modified. What it is known so far is that 80% of Cys34 is in a form that is not reducible by 2-mercaptoethanol or DTT, even under denaturing conditions. Some possible modifications were evaluated but their signals were not found in none of the MS obtained. Nevertheless, the Cys34 peptide is easily observed when performing mass spectrometry experiments. In fact, over the years we have observed this peptide with different modifications [8, 9]. Further studies need to be done in order to shed light on the nature and identity of the modification.

Regarding the reaction between DomI and DTNB, the rate constant was determined to be 110.3 $M^{-1} s^{-1}$. This value is ~24 times higher than that with HSA (4.6 $M^{-1} s^{-1}$). A combination of factors can explain the increased reactivity of DomI Cys34. First, the binding of ligands is known to induce conformational changes in the protein [25, 36–40]. For example, when the ligand is a fatty acid, conformational changes result in the opening of the crevice where Cys34 is located [25, 39, 40]. As a consequence, the exposure and solvent accessibility of the thiol increase [25, 41, 42] and the p$K_a$ decreases ~ 0.5 pH units [25]. This results, for example, in a ~

6-fold increase in the rate constant of the reaction between DTNB and 5/1 fatty acid-HSA complex (*e.g.* lauric, myristic, palmitic, stearic and oleic acids) compared to delipidated HSA [25]. To clarify whether the binding of ligands was responsible for increased reactivity of DomI and DTNB, we tried to delipidate DomI with activated charcoal as indicated for HSA [21]. However, this was not possible because DomI precipitated at pH lower than 4.7. Second, the absence of domains II and III could imply a reduction in the number of non-covalent interactions, leading to changes in DomI tertiary structure with respect to HSA. These changes could affect amino acids that are active players in the determination of the reactivity of Cys34 [10, 30]. Third, a decrease in the total negative charge of DomI with respect to HSA could also accelerate the reaction with the negatively charged DTNB.

In summary, in this work HSA DomI was expressed in *P. pastoris* system and successfully purified from contaminant proteins and a brown pigment. The thiol of Cys34 was partially reducible and its reactivity towards the disulfide DTNB was studied and compared to that of HSA. These results open the door to new experiments aiming to fully characterize the properties of the oxidative derivatives of HSA, including the generation of mutants in key residues and nuclear magnetic resonance structural studies. This is of relevance as, up to now, there is only one deposited structure of HSA in the sulfenic acid form (PDB ID 5Z0B), and none in further oxidation states such as sulfinic and sulfonic acid.

## Supporting information

**S1 Raw images.**
(PDF)

## Acknowledgments

We are grateful to Drs. Mariana Bonilla, Bruno Manta, Sonia Rodriguez, Mario Señorale and Andrea Villarino (Universidad de la República, Uruguay) for helpful discussions.

## Author Contributions

**Conceptualization:** Lucía Turell.

**Funding acquisition:** Lucía Turell.

**Investigation:** Martina Steglich, Rodrigo Lombide, Ignacio López, Lucía Turell.

**Methodology:** Ignacio López, Madelón Portela, Martín Fló, Mónica Marín.

**Resources:** Madelón Portela, Martín Fló, Mónica Marín, Beatriz Alvarez.

**Supervision:** Beatriz Alvarez, Lucía Turell.

**Visualization:** Martina Steglich.

**Writing – original draft:** Martina Steglich.

**Writing – review & editing:** Ignacio López, Martín Fló, Mónica Marín, Beatriz Alvarez, Lucía Turell.

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
