## [Decision Letter · Decision Letter 0]

7 Sep 2020

PONE-D-20-25038

Expression, purification and initial characterization of human serum albumin domain I and its cysteine 34

PLOS ONE

Dear Dr. Turell,

Thank you for submitting your manuscript to PLOS ONE. After careful consideration, we feel that it has merit but does not fully meet PLOS ONE’s publication criteria as it currently stands. Therefore, we invite you to submit a revised version of the manuscript that addresses the points raised during the review process.

ACADEMIC EDITOR: Please try to improve and correct your manuscript according to the reviewers' criticism.

We look forward to receiving your revised manuscript.

Kind regards,

Eugene A. Permyakov, Ph.D., Dr.Sci.

Academic Editor

PLOS ONE

Journal Requirements:

2.PLOS ONE now requires that authors provide the original uncropped and unadjusted images underlying all blot or gel results reported in a submission’s figures or Supporting Information files. This policy and the journal’s other requirements for blot/gel reporting and figure preparation are described in detail at https://journals.plos.org/plosone/s/figures#loc-blot-and-gel-reporting-requirements and https://journals.plos.org/plosone/s/figures#loc-preparing-figures-from-image-files. When you submit your revised manuscript, please ensure that your figures adhere fully to these guidelines and provide the original underlying images for all blot or gel data reported in your submission. See the following link for instructions on providing the original image data: https://journals.plos.org/plosone/s/figures#loc-original-images-for-blots-and-gels.

Reviewers' comments:

Reviewer's Responses to Questions

**Comments to the Author**

1. Is the manuscript technically sound, and do the data support the conclusions?

Reviewer #1: Yes

Reviewer #2: Partly

2. Has the statistical analysis been performed appropriately and rigorously? 

Reviewer #1: N/A

Reviewer #2: N/A

3. Have the authors made all data underlying the findings in their manuscript fully available?

Reviewer #1: Yes

Reviewer #2: Yes

4. Is the manuscript presented in an intelligible fashion and written in standard English?

Reviewer #1: Yes

Reviewer #2: Yes

5. Review Comments to the Author

Reviewer #1: The manuscript reports the expression of human serum albumin (HSA) domain 1 in the commercially available Pichia pastoris expression system, and determination of reactivity of the sole free cysteine at amino acid position 34 (Cys-34). The study showed that the thiol of Cys-34 in the recombinant domain 1 protein had a 24-fold increased rate constant as compared with the full sized albumin protein.

The recombinant protein expression of domain 1 of HSA was reported previously and is commercially available at https://www.albuminbio.com/. Therefore, this study lacks originality and novel findings. Although the methods used in the study are sound and fair, the results reported in this study are not new and are a simple repetition of previously published studies.

Reviewer #2: Re: Expression, purification and initial characterization of human serum albumin domain I and its cysteine 34

The submitted manuscript thoroughly describes purification of a human serum albumin (HSA) domain containing single free cysteine residue. Then authors carefully investigated possible chemical states of this cysteine using MALDI MS and chemical kinetics. HSA is a major protein of human plasma, and its free cysteine supposedly actively participates in the redox reactions.

Despite overall positive impression from the article, I cannot recommend it for publication in its current state due to many inconsistencies both in the kinetics and MS parts of the investigation.

Major comments

1. Ln. 308 and Fig. 5 (inset): A normal isotopic distribution will not have a frequency peak for isotopic variants taller than the peak for average mass (since for biogenic elements all heavier isotopes have lower abundancy). The high peak with m/z 2491 indicates to me that the carboxymethylation did take place.

2. Ln. 324 and Fig. 6B: From the figure it’s not clear why the authors decided to wander off into some complicated fitting. Both black and blue curves will be adequately described by a single pseudo-first order equation.

3. Fig. 6C. This is a major flaw of the article: from basic chemical kinetics, the reaction rates do depend on the concentrations, but their constants do not. The authors need to review their data and calculations and present them in a correct form.

4. Ln. 329 and throughout: the authors could not possibly determine the second order constants, because the concentration of one of the reagents was kept constant.

5. Ln. 345: Its not quite clear how thiol/DomI ratio was calculated.

Minor and technical comments

1. Ln. 142-148: It’s not clear how the protein concentration was calculated, from the calibration line or from the theoretical extinction coefficient.

6. PLOS authors have the option to publish the peer review history of their article (what does this mean?). If published, this will include your full peer review and any attached files.

Reviewer #1: No

Reviewer #2: No

---

## [Author Response · Author response to Decision Letter 0]

29 Sep 2020

Answer to reviewers

We thank the reviewers for the positive and constructive comments. We have considered all the suggestions and have performed a thorough revision of the manuscript.

Reviewer 1

The recombinant protein expression of domain 1 of HSA was reported previously and is commercially available at https://www.albuminbio.com/. Therefore, this study lacks originality and novel findings. Although the methods used in the study are sound and fair, the results reported in this study are not new and are a simple repetition of previously published studies.

Answer: We agree with the reviewer that the recombinant production of HSA DomI is not novel. In fact, a previous work where the three HSA domains were produced in P. pastoris was cited in this manuscript (Ref #14, Dockal M, et al, JBC, 1999). However, the main results of the manuscript are new and have not been published before. First, valuable data on how to purify proteins from pigments produced by P. pastoris, a topic about which there is scarce information in the literature, is presented. Second, studies on the redox status of the only free cysteine of DomI are reported. The redox state of this residue is sometimes overlooked despite the fact that it can affect the secondary structure and net charge of the protein (Bruschi, et al, BBA, 2013). It was confirmed that, even under the expression conditions (96 h, 30°C, agitation), a fraction of Cys34 was still reducible. Furthermore, experimental conditions to reduce it selectively were found, avoiding the reduction of the internal disulfides of the protein. Last, the reactivity of the thiol of DomI with DTNB was evaluated and compared with that of HSA. Therefore, we think our manuscript may be of interest not only within the field of albumin research but also within the areas of thiol redox biochemistry and of expression systems for the production of recombinant proteins. 

Reviewer 2 

Major comments 

1. Ln. 308 and Fig. 5 (inset): A normal isotopic distribution will not have a frequency peak for isotopic variants taller than the peak for average mass (since for biogenic elements all heavier isotopes have lower abundancy). The high peak with m/z 2491 indicates to me that the carboxymethylation did take place.

Answer: We agree that for small peptides the probability of possessing one heavier isotope, like 13C, is low. Thus, the monoisotopic peak is expected to be the tallest. However, as the peptide size increases, the probability of having heavier isotopes increases as well. This is why in a peptide of m/z 2491, it is expected to find peaks taller than the one corresponding to the monoisotopic mass. 

To illustrate on this, we plotted the isotopic distribution of the Cys34 peptide when only iodoacetic acid labeling was performed, as described in Fig. 4 (included in the new version of the manuscript as an inset of Fig. 4A). The proportions of the peaks are the same as in Fig. 5A inset validating our conclusion that the isotopic distribution shown in this figure corresponds to the Cys34 peptide, carbamidomethylated only and not carboxymethylated. 

2. Ln. 324 and Fig. 6B: From the figure it’s not clear why the authors decided to wander off into some complicated fitting. Both black and blue curves will be adequately described by a single pseudo-first order equation.

Answer: We agree with the reviewer that the data of a pseudo-first order reaction should fit to a simple exponential function. However, in our case, the data did not fit to a single exponential equation but to a single exponential plus a straight line equation. This is due to the existence of a second and slower reaction (linear part of the fit), assigned to secondary reactions of DTNB like dismutation. The first phase (exponential fit) corresponds to the reaction of DTNB and the thiol of DomI, thus the observed exponential rate constants (kobs) were obtained exclusively from the exponential part of the fit. 

For clarity, the text was modified as follows:

“Purified DomI was incubated with 2-mercaptoethanol (10 mM, 30 min, room temperature), excess reductant was eliminated by gel filtration and the resulting reduced DomI (11.5 µM) was mixed with increasing concentrations of DTNB in pseudo-first order excess (50-200 µM). For comparison, the experiment was also performed with reduced HSA (20 µM). The time courses showed that DTNB reacted faster with DomI than with HSA (Fig 6B). Exponential plus straight line equations were fitted to the data. The first phase (exponential fit) corresponds to the reaction between DTNB and Cys34, and the second phase (straight line fit) can be explained in terms of secondary reactions such as DTNB dismutation [24]. The amplitude and pseudo-first order rate constant (kobs) were obtained. Then, kobs were plotted against DTNB concentration (Fig 6C). From this secondary plot, second order rate constants of 110.3 M-1 s-1 and 4.6 M-1 s-1 were determined for DomI and HSA, respectively (20 mM phosphate, pH 7.4, 25 ºC). Thus, it can be concluded that DomI Cys34 reacted with DTNB ~24 times faster than HSA Cys34. A control of non-reduced DomI (11.5 µM) and DTNB (200 µM) was also performed and no reaction was observed (not shown).”

3. Fig. 6C. This is a major flaw of the article: from basic chemical kinetics, the reaction rates do depend on the concentrations, but their constants do not. The authors need to review their data and calculations and present them in a correct form.

Answer: We respectfully disagree with the reviewer. Our kinetic analysis is flawless. Please see the answer to the next question. 

4. Ln. 329 and throughout: the authors could not possibly determine the second order constants, because the concentration of one of the reagents was kept constant.

Answer: We completely agree with the reviewer that reaction rates depend on the concentrations but second order rate constants do not. In our experiment, the reaction followed pseudo-first order kinetics because DTNB was in excess over DomI. We performed the experiment at different DTNB concentrations and from the exponential plus straight line fit of the data presented on Fig. 6B, we obtained the observed exponential rate constant (kobs) at each DTNB concentration. As expected for pseudo-first order kinetics, the kobs is related to the second order rate constant (k) according to the equation kobs = k [DTNB]. Thus, the second order rate constant was obtained from the slope of linear fit of the kobs versus [DTNB] plot. This is what it is shown in Fig. 6C.

5. Ln. 345: Its not quite clear how thiol/DomI ratio was calculated.

Answer: We thank the reviewer for this comment. For clarity, the text was modified as follows:

“The concentration of reduced thiols in DomI was determined from the amplitude of the time courses of the reaction with DTNB. To account for the fact that the first ~10 seconds were missed, the amplitude obtained from the fit was corrected considering as the initial absorbance, the sum of the absorbance of the controls (only DTNB and only DomI) (Fig 6B). Taking into account the total protein concentration (DomI) used in the experiment, a thiol/DomI ratio of 0.20 was obtained. This suggests that 20% of DomI Cys34 was reduced after 2-mercaptoethanol incubation, 30 min at room temperature. Similar results were obtained with overnight reduction at 4 ºC (not shown).”

Minor and technical comments 

1. Ln. 142-148: It’s not clear how the protein concentration was calculated, from the calibration line or from the theoretical extinction coefficient.

Answer: We thank the reviewer for this comment. For clarity the text was modified as follows:

“Protein quantification

Three methods were evaluated to determine DomI concentration. Bradford, bicinchoninic acid assay (BCA) and 280 nm absorbance determinations were performed [22,23]. For Bradford and BCA measurements, calibration curves were done with delipidated HSA. For the BCA assay, samples were heated at 60 °C during 15 min for color development. Controls with the sample in the absence of BCA or Bradford reagents were included. For protein quantification by 280 nm absorbance, an extinction coefficient of ε = 11055 M-1 cm-1 was determined from the primary sequence of DomI using the ProtParam tool (ExPASy, Bioinformatics Resource Portal).” 

And, 

“DomI expression and purification

Regarding protein quantification, the presence of the pigment, mainly in the dialyzed sample, did not allow the direct measurement of the absorbance at 280 nm. Also, interferences due to the pigment were observed with the BCA method (data not shown). Thus, protein concentration was determined using the Bradford assay both in the dialyzed and purified DomI samples. An estimated yield of 0.1 mg of DomI per mL of culture was obtained.”

---

## [Editor Report · Decision Letter 1]

30 Sep 2020

Expression, purification and initial characterization of human serum albumin domain I and its cysteine 34

PONE-D-20-25038R1

Dear Dr. Turell,

We’re pleased to inform you that your manuscript has been judged scientifically suitable for publication and will be formally accepted for publication once it meets all outstanding technical requirements.

Kind regards,

Eugene A. Permyakov, Ph.D., Dr.Sci.

Academic Editor

PLOS ONE
---

## [Editor Report · Acceptance letter]

2 Oct 2020

PONE-D-20-25038R1 

Expression, purification and initial characterization of human serum albumin domain I and its cysteine 34 

Dear Dr. Turell:

I'm pleased to inform you that your manuscript has been deemed suitable for publication in PLOS ONE. Congratulations! Your manuscript is now with our production department. 

Kind regards, 

on behalf of

Prof. Eugene A. Permyakov 

Academic Editor

PLOS ONE